# First Report and Characterization of a Plasmid-Encoded *bla*_SFO-1_ in a Multi-Drug-Resistant *Aeromonas hydrophila* Clinical Isolate

**DOI:** 10.3390/microorganisms12030494

**Published:** 2024-02-29

**Authors:** Troy Skwor, Dan Christopher Jones, Caitlin Cahak, Ryan J. Newton

**Affiliations:** 1School of Biomedical Sciences and Health Care Administration, University of Wisconsin-Milwaukee, Milwaukee, WI 53211, USA; dcjones@uwm.edu; 2Wisconsin Diagnostics Laboratory, Milwaukee, WI 53226, USA; 3School of Freshwater Sciences, University of Wisconsin-Milwaukee, Milwaukee, WI 53204, USA; newtonr@uwm.edu

**Keywords:** SFO-1, *Aeromonas*, antimicrobial resistance, carbapenemase, extended-spectrum beta-lactamases (ESBLs)

## Abstract

Antibiotic resistance remains one of the most pressing public health issues facing the world today. At the forefront of this battle lies the ever-increasing identification of extended-spectrum beta-lactamases and carbapenemases within human pathogens, conferring resistance towards broad-spectrum and last-resort antimicrobials. This study was prompted due to the identification of a pathogenic *Aeromonas hydrophila* isolate (strain MAH-4) collected from abdominal fluid, which presented a robust resistance pattern against second-, third-, and fourth-generation cephalosporins, ertapenem, ciprofloxacin, gentamicin, levofloxacin and moxifloxacin, and beta lactam/beta-lactamase inhibitor combinations. Whole genome sequencing was performed and identified a 328 kb plasmid (pMAH4) encoding 10 antibiotic resistance genes, including *bla*_SFO-1_, *bla*_TEM-1_, and *bla*_OXA-1_ of *A. hydrophia* MAH-4. This is the first report of beta-lactamase SFO-1 within a clinical strain of *Aeromonas*. Due to the remarkable sequence identity of pMAH4 to plasmids associated with *Enterobacterales* genera like *Klebsiella* and the extensive capabilities of *Aeromonas* for horizontal gene transfer, our identification of a clinical isolate encoding SFO-1 on a plasmid suggests antibiotic resistance gene mobility between *Enterobacterales* and non-*Enterobacterales* species.

## 1. Introduction

Antimicrobial resistance (AMR) remains one of the world’s top public health emergencies, claiming roughly 1.27 million lives in 2019 [1]. With discoveries of new AMR far outpacing discoveries of new antibiotics, this situation is only becoming more dire. Projections currently estimate that AMR bacterial-caused mortalities will reach 10 million by the year 2050 [1]. Despite overwhelming evidence and repeated warnings surrounding this public health crisis, global economic development and increasing access to medical treatments have continued to drive an increase in antimicrobial consumption [2]. Alarmingly, broad-spectrum and last-resort antibiotics are becoming more frequently employed than ever before [2]. Consequently, as these compounds’ usage increases, so does the identification of bacterial strains resistant to them [3]. This often leads to longer hospital stays, higher healthcare costs, and increased mortality, particularly within areas with developing economies [4]. Within bacterial pathogens most responsible for human mortalities, extended-spectrum beta-lactamases (ESBLs) and carbapenemases rank among the top acquired antibiotic resistance genes (ARGs) [1]. AMR among *Enterobacterales* members *Escherichia coli* and *Klebsiella pneumoniae* alone were solely responsible for 23.4% and 19.9% of deaths attributed to AMR pathogens in high- and low-income regions, respectively [1]. Therefore, the Center for Disease Control and Prevention has deemed members of carbapenem-resistant and ESBL-producing *Enterobacterales* as urgent and serious threats to human health [5].

Some of the most pervasive ESBLs currently being identified within clinical pathogens originate from common broad-spectrum beta-lactamases, such as TEM, SHV, OXA, and CTX-M [6], with the CTX-M type demonstrating the greatest prevalence worldwide [7]. Due to their commonality, as well as the high degree of genetic diversity within the CTX-M family, their dissemination throughout the world has often been referred to as the “CTX-M pandemic” [8]. While these beta-lactamases and ESBLs constitute the majority of ARGs within common pathogens, other concerning ESBLs are being identified more frequently than they were in years past [7]. SFO-1, regulated by the AmpR regulator, is a rarer class A ESBL with the capacity to hydrolyze numerous beta-lactams other than cephamycins and carbapenems. Initially identified in 1988 from a self-transferable plasmid within a clinical strain of *Enterobacter cloacae* [9], SFO-1 has since been found within clinical isolates of various species of *Enterobacterales* such as *Klebsiella* spp., *Escherichia coli*, and *Enterobacter cloacae* complex [7]. The first SFO-1 epidemic was identified in Spain and was caused by *Enterobacter cloacae* between 2006–2009 [10]. The SFO-1 outbreak was significantly correlated with administration of beta-lactam antibiotics, chronic renal failure, tracheostomy, and prior hospitalization, further emphasizing the impact of antibiotic overconsumption [10]. Then, between 2011–2015, SFO-1-encoding strains of *E. hormaechei* evolved into epidemic clones (ST93, ST114, and ST418) among community-acquired strains found in four provinces in China [7].

Apart from minimizing antibiotic overconsumption, the search for a solution to the growing AMR issue must also consider the horizontal spread of ARGs between various biome resistomes [11]. To confront challenges on multiple fronts like what is seen within the issue of AMR, approaches such as One Health use the expertise from several disciplines to identify solutions to complex issues facing the world [5]. However, a challenge in using this tactic against AMR resides in identifying a proper indicator species. One potential indicator which has been proposed is *Aeromonas* [3]. This genus of bacteria resides ubiquitously in aquatic environments across the globe, including fresh and brackish waters, and has been correlated with a variety of diseases among cold- and warm-blooded animals [12]. *Aeromonas*-related diseases within humans often afflict children or those with compromised immune systems. Predominantly, these diseases include gastrointestinal infections like gastroenteritis or soft-tissue infections such as cellulitis, abscesses, and necrotizing fasciitis [13,14]. Furthermore, aeromonads readily participate in both intra- and interspecies horizontal gene transfer (HGT) [15], allowing for monitoring of ARGs within an environment [16]. Aquaculture and wastewater treatment facilities, both common habitats for *Aeromonas* [3,17], have been identified as hot-spots of ARG transmission [18]. Therefore, it has been proposed that aeromonads may provide a useful tool in tracking antimicrobial resistance patterns around the world [3,19].

Here, we report and characterize the first clinical case of plasmid-encoded *bla*_SFO-1_ in a multi-drug-resistant isolate of *Aeromonas hydrophila* to further understand the driving factors of resistance, especially among ESBLs and carbapenemases.

## 2. Methods

### 2.1. Bacterial Isolation and Identification

This study was approved on 18 December 2019 by the Medical College of Wisconsin and the Froedtert Institutional Review Board (PRO# 00036609). Abdominal fluid was surgically collected from a patient and submitted to the hospital’s diagnostic laboratory in a sterile syringe. The sample was submitted and worked up as an aerobic/anaerobic culture. Confirmation of the isolates was performed using a Bruker MALDI-TOF analyzer (Bruker Daltonics, Bremen, Germany) with subsequent identification comparing the protein spectrum of the isolate to the Bruker MTB Compass Library (Revision H, 10,833 MSP). Glycerol frozen stocks were made for future analysis.

### 2.2. Antimicrobial Susceptibility

A fresh subculture of the clinical isolate was used to perform antimicrobial susceptibility testing using a BD Phoenix M50 analyzer and BD EpiCenter System V7.21 (Becton, Dickinson and Company, Sparks, MD, USA). The BD EpiCenter System uses current CLSI guidelines and customized rules designated by the laboratory to determine the susceptibility interpretations from the MIC results given from the BD Phoenix. A Kirby disk diffusion assay was used to also analyze antimicrobial susceptibility to the following antibiotics: aztreonam (30 mcg), cefotaxime (30 mcg), chloramphenicol (30 mcg), ciprofloxacin (5 mcg), meropenem (10 mcg), tetracycline (30 mcg), sulfamethoxazole–trimethoprim (23.75/1.25 mcg). Antimicrobial susceptibility was determined using values provided in the Clinical and Laboratory Standards Institute M45-A for *Aeromonas* spp. [20].

### 2.3. Isolation of Genome DNA and Whole Genome Sequencing

An overnight culture of *A. hydrophila* MAH-4 was pelleted and genomic DNA was obtained using a Zymo DNA Miniprep (Zymo Research, Irvine, CA, USA). Long reads and short paired-end reads (2 × 151 bps) were obtained by the Oxford Nanopore (Oxford Nanopore, Oxford, UK) platform and Illumina NextSeq 2000 (Illumina, Inc., San Diego, CA, USA) platform, respectively, for whole genome sequencing (WGS). Quality control and adapter trimming were performed using bcl-convert and porechop followed by hybrid assembly of the reads using Unicycler [21]. The genome was submitted to NCBI Accession no. CP143514 and CP143515 (pMAH4). Annotations of the assemblies were performed using BV-BRC 3.33.16 [22]. Phylogenetic reconstruction of MAH-4 with other representative *Aeromonas* species was carried out using 100 different conserved and concatenated genes and was run in BV-BRC 3.33.16 (Appendix A) [22]. Further confirmation of species was performed by determining an average nucleotide identity (ANI) in comparison to reference strain *Aeromonas hydrophila* subspecies hydrophila ATCC 7966 using JSpeciesWS version 4.1.1 [23].

### 2.4. Bioinformatics

We conducted a blastn analysis [24] against the NCBI nr nucleotide database to identify associations, i.e., contiguous regions of high sequence identity, of the pMAH4 DNA sequence with other bacterial genomes (accessed on 18 January 2024). Seven bacterial isolate genomes with very high identity (>99.9%) and ≥18% coverage of pMAH4 were selected for further analysis and visualization. Additionally, the blastn analysis identified seven *Aeromonas* spp. genomes in NCBI that had high identity (>95%) to a portion (>5%) of pMAH4. The matching sequence region from the 14 selected genomes was added to the program Blast Ring Image Generator (BRIG; [25]) for sequence identity visualization and mapping. Within BRIG, pMAH4 was set as the “reference sequence” and the other 14 genomes as “query sequences”. Nucleotide sequences were used in the pairwise blastn query between all 14 sequences and pMAH4, and BLAST settings were left as default. The final BRIG image was exported, and final image annotations were modified in Adobe Illustrator 2024 (Adobe Inc., San Jose, CA, USA).

Various online bioinformatic tools were used to further characterize pMAH4. Identification of antibiotic resistance genes (ARGs) was conducted in both Proksee CARD [26,27] and ResFinder 4.4.2 [28] from the Center for Genomic Epidemiology. The ID threshold was set at 90% and at 80% for minimum length. BLAST^®^ was further used to analyze ARGs on the chromosome because it has a larger library of *Aeromonas* sequences. Identification and location of mobile genetic elements (MGEs) were performed using MobileElementFinder [29], ISFinder [30], and Proksee mobileOG-db [31]. PHASTER was used to identify similarity to bacteriophage DNA [32,33,34]. Comparison of *tcb* operon nucleotide similarity among members of the Aeromonadaceae/Succinivibrionaceae group (taxid: 135,624) was performed with BLAST^®^ [35].

## 3. Results and Discussion

### 3.1. Antimicrobial Susceptibility Profile of a Clinical Aeromonas *sp.*

A clinical isolate was acquired from the abdominal fluid of a patient with a perforated abdominal ulcer and identified using MALD-TOF as an *Aeromonas hydrophila/veronii/jandaei*. It was designated as strain MAH-4. *Aeromonas* sp. MAH-4 was resistant to second- (i.e., cefoxitin and cefuroxime), third- (ceftriaxone and ceftazidime), and fourth-generation (i.e., cefepime) cephalosporins, ertapenem, ciprofloxacin, gentamicin, levofloxacin and moxifloxacin, and beta-lactam/beta-lactamase inhibitor combinations (Table 1). However, this isolate was susceptible to tetracycline and trimethoprim–sulfamethoxazole (Table 1).

### 3.2. Whole Genome Sequence Analysis

Due to the intense antimicrobial resistance patterns, we wanted to understand the molecular mechanisms driving these resistance phenotypes. Therefore, whole genome sequencing of strain MAH-4 was performed with both long and short reads resulting in two closed contig loops, a chromosomal genome of 5,110,450 bp with a 327,540 bp plasmid (pMAH4). The chromosome was 60.78% GC content with an estimated 4,803 CDS (Figure 1). Phylogenetic analysis using 100 different genes comprising 115,401 concatenated nucleotides placed strain MAH-4 within a clade of *A. hydrophila* species (Figure 2). The MAH-4 chromosome also had 96.61% average nucleotide identity (ANI) with *A. hydrophila* subspecies ATCC 7966 which is above the 95% cutoff value for grouping genomes within the same species. From the genome-based phylogeny and ANI results, we concluded that strain MAH-4 is an *Aeromonas hydrophila* species. *A. hydrophila* has been commonly associated with a myriad of clinical diseases ranging from gastroenteritis and colitis [36] to wound infections and life-threatening necrotizing fasciitis [36]. The evolution of many clinically relevant *Aeromonas* species has included the acquisition of multiple beta-lactamases resulting in intrinsic resistance to many beta-lactam antibiotics. Often, the chromosome of these organisms carries Ambler class B2 metallo-b-lactamase, Ambler class D penicillinase, and Ambler class C cephalosporinases [37]. *A. hydrophila* MAH-4 carries all of these beta-lactamases: *bla*_cphA2_, *bla*_OXA-12_, and *bla*_ampC_ (Table 2). Additionally, *A. hydrophila* MAH-4 encodes multiple genes associated with the ATP-binding cassette family (ABC), the resistance-nodulation-cell division (RND) family, the small multi-drug-resistance (SMR) family, the multi-drug and toxic compound extrusion family (MATE), and tri- and tetrapartite multi-drug efflux pumps. These pumps may enhance or confer resistance patterns observed beyond the presence of ARGs [38].

### 3.3. Characterization of pMAH4

To further our understanding of the transmission of antimicrobial resistance, we characterized the genetic signatures associated with resistance of *A. hydrophila* MAH-4. The 327,540 bp plasmid (pMAH4) had a 52.89% G+C content but could not be assigned an incompatibility group using PlasmidFinder [39], although a *traX* gene contained 100% identity to TraX of the IncF plasmid (Figure 3). Ten different ARGs intermixed with numerous MGE were encoded on this plasmid, creating a mosaic evident of multiple mobilization events. Beyond the intrinsic resistance provided by the chromosomally encoded beta-lactamases, *A. hydrophila* pMAH4 encoded *bla*_SFO-1_, *bla*_TEM-1_, and *bla*_OXA-1_ (Figure 3 and Table 2)_._ This is the first report of *bla*_SFO-1_ from a clinical strain of *Aeromonas.*

SFO-1 was first identified in an *Enterobacter cloacae* isolate in Japan [9], which shared strong amino acid similarity to chromosomal-encoded AmpA in *Serratia fonticola.* Typically, SFO-1 is identified in members of the *Enterobacterales* and is located on plasmids [7,10]. However, to the authors’ knowledge, this is the first clinical case of an *Aeromonas* species or non-*Enterobacterales* encoding SFO-1. Immediately adjacent to the *bla_SFO-1_* was the transcriptional regulator *ampR* (Figure 1). SFO-1 production provides hydrolysis to oxyimino-cephalosporins like cefotaxime, although this activity is inhibited in the presence of imipenem or clavulanic acid [9]. BLASTp only identified one isolate among *Aeromonas* species for SFO-1 in the NCBI nt database and it was *Aeromonas hydrophila* AFG_SD03_1510_Ahy_093 from stray dog fecal matter in Afghanistan [40]. Outside of this one isolate, all other SFO-1-encoding organisms listed in the NCBI Identical Protein Groups (n = 515) were from *Enterobacterales* members. Although, it is possible that the frequency of SFO in clinical isolates may be underreported due to misidentification as CTX-M in lateral flow assays [41].

Upstream of the *bla_SFO-1_*, the aminoglycosides phosphotransferase *aph(6)-Id* and *aph(3′)-lb* are found as part of a Tn3 family including the transposase (TnpA) and resolvase (TnpR). Additionally, the presence of a composite transposon Tn6082 and insertion sequence ISAhy2 upstream of *bla_SFO-1_-ampR* followed by ISCfr1 and Tn3 transposon downstream suggests that numerous recombination events shaped the content of pMAH4. Downstream of the SFO-1-ampR is a class 1 composite transposon flanked by IS6100 elements rich in ARGs between 30,939–53,611 bp: IS6100-*mphR(A)-mrx-mphA*-IS26- *aac(3)-IId* -ISCfr1-*bla_TEM_-trpA-trpR-intI1- aac(6′)-lb-cr6-bla_OXA1_-catB3-qacEdelta1-sul1*-IS6100 (Figure 3). The presence of the macrolide 2′-phosphotransferase I (mphA) alongside mrx and mphR in a composite plasmid was first described in *A. hydrophila* from pigs in Oklahoma, USA, in 2006 [42]. This gene combination, which provides macrolide resistance, is evident in *Aeromonas hydrophila* within both the chromosome and plasmids, supporting a high degree of mobility between the two [26]. Directly downstream of the macrolide ARGs is an IS26 element followed by the aminoglycoside acetyltransferase aac(3)-IId, which has also been found on both the chromosome and plasmids from *A. hydrophila* [26].

Two of the earliest identified ESBLs associated with plasmids were identified on pMAH4: TEM-1 and OXA-1. The ESBL TEM-1 has been around since the 1960s and provides resistance to first-generation cephalosporins, while OXA-1 enables hydrolysis of fourth-generation cephalosporins, like cefepime [8]. The presence of an aminoglycoside *N*-acetyltransferase (aac(6′)-lb-cr) is commonly found in class 1 integrons and associated with IS26 elements similar to pMAH4 [43]. This ARG provides resistance to aminoglycosides and moderate resistance due to fluoroquinolone acetylation [44] and has been found among ciprofloxacin-resistant *Aeromonas hydrophila* clinical isolates as well as within environmental samples surrounding aquaculture and hospitals [45]. Lastly, *catB3* and *sul1* were found furthest downstream of the composite transposon. Chloramphenicol resistance is associated with the inducible production of chloramphenicol O-acetyltransferase (*catB3*) and has been identified in many species of *Aeromonas*, including on both a plasmid and chromosome of *A. hydrophila* [26]. The last ARG, a sulfonamide-resistant dihydropteroate synthase (sul1), is commonly found among Gram-negative bacteria, including *Aeromonas* species where it is more common to be encoded on the chromosome than the plasmid [26]. However, it is a common ARG among the *Aeromonas* mobilome and in some cases has been identified on plasmids and associated with transposons, and class 1 integrons, as well as clustered within rich ARG regions [46]. The absence of resistance to SXT in the presence of *sul1* might be associated with a decreased expression considering there is a seven-nucleotide overlap of *qacE* delta 1 [47].

Although a large portion of pMAH4 contains unknown open reading frames (ORFs), a mercury resistance operon (MerRTPEDAFPTR) encoding genes to detoxify the heavy metal mercury is present. MerT, P, F, and E are associated with transport, whereby MerD serves as a transcriptional repressor. This operon has been identified in both Gram-positive and Gram-negative bacteria. Considering the ubiquitous presence of *Aeromonas* in aquatic environments and common mercury contaminants in anthropogenic-impacted waters [8], the acquisition of a mercury resistance operon is not surprising. Additionally, conjugative transfer of plasmids increases in the presence of low levels of mercury contamination [48] and this operon has already been identified in clinical *Aeromonas* strains showing varying resistance to mercury [49].

### 3.4. Evolution of pMAH4

Comparing pMAH4 to most other plasmids using BLAST provided some insights into its potential origin. The majority (75%) of pMAH4 was made up of DNA with very few predicted ORFs and an absence of high sequence identity (>50%) to other sequences in the NCBI nucleotide database (Figure 4 and Appendix A). The lone exception was a very high identity match to an *A. caviae* isolate acquired from a hospital patient’s urine in November 2022 from the province of Guangdong in China. This isolate’s genome contained a plasmid pAC1520 with 95.88% nucleotide identity to pMAH4 within this region, but the remaining ~25% (75 kb) of the pMAH4 plasmid did not match pAC1520. This region of pMAH4 was rich in ARGs and MGEs and contained segments with a very high sequence identity (often >99.9%) to plasmids from members of *Enterobacterales*, predominantly from the genera *Klebsiella* and *Enterobacter* (Figure 4 and Appendix A). These isolates were acquired from human samples from at least four different provinces in China from 2016–2021 and shared a very high sequence identity (99.9–100%) to the ARG-containing region on pMAH4 including *bla_SFO-1_
*(Figure 4 and Appendix A)_._

Among *Aeromonas* spp., there were seven genomes that contained a very high sequence identity (99.9–100%) to the plasmid region containing the ten ARGs (Figure 4 and Appendix A). Only one genome, *Aeromonas hydrophila* AFG_SD03_1510_Ahy_093, harbored *bla_SFO-1_*. This strain of *A. hydrophila* was acquired from stray dog feces in Afghanistan in 2015 [40]. Although it was the only *Aeromonas* isolate encoding a *bla_SFO-1_*, it was missing the majority of the other ARGs encoded downstream of *bla_SFO-1_* in pMAH4: *mphA-aac(3)-IId-bla_TEM-1_-aac(6′)-lb-cr-bla_OXA-1_-catB3-qacEdelta1-sul1*. Only one other *Aeromonas* isolate had high sequence identity to this ARG-rich region, *Aeromonas caviae* SCLZ552. This isolate was acquired from wastewater in the Sichuan province in China in 2019. One other isolate, *Aeromonas caviae* WP3-S18-ESBL-02, from Tokyo, Japan, had high plasmid sequence identity to the ARG-rich region in pMAH4, but it was broken into several smaller matching regions. This *A. caviae* isolate was acquired in 2018 from wastewater effluent as well. *Aeromonas* spp. are common municipal and hospital wastewater residents and also comprise a high percentage of ESBL-producing bacteria in final treated wastewater effluents, even post-disinfection [50]. *Aeromonas* spp., discharged in final effluent, have been identified with the following carbapenemases: *bla*KPC-2, *bla*VIM-2, *bla*OXA-48, *bla*IMP-13, *bla*GES, and *bla*MOX genes [51]. Their presence in treated effluent emphasizes the potential role *Aeromonas* can serve as a vehicle for transferring dangerous ARGs across environmental sectors [3] as well as an opportunist for exchanging ARGs both intra- and interspecies.

Other *Aeromonas* isolates with a high degree of nucleotide similarity to pMAH4, though without *bla_SFO-1_*, were from clinical strains. These isolates spanned three different Chinese provinces from 2019–2022 ranging from urine (*A. caviae* AC1520) to bile (*A. caviae* FAHZZU2447). One study performed between 2012–2016 by the NIH Clinical Center in the USA focused on identifying potential reservoirs of carbapenemase-producing bacteria around hospital environments to help understand the cause of increased carbapenem-resistant hospital-acquired infections [52]. The study identified six *bla*_KPC_-encoding *Aeromonas* isolates from within wastewater manholes associated with hospitals. In this study, one patient did have an *Aeromonas* strain containing a *bla*_KPC-2_ on a 143.4 kb plasmid (pASP-135 from *A. hydrophila* AHNIH1) though it was unrelated to the hospital environmental isolates [52]. Many of these isolates encoded a chromosomal *trb* operon, which is associated with type IV secretion systems [53]. *A. hydrophila* MAH-4 also encodes a *trb* operon (trbK-VirD4-trbBCDEJKLFGI) in its chromosome. This region shared the highest nucleotide identity (≥99.98%) with the *trb* operon in 244 kb and 49 kb *bla*_GES_-encoding plasmids from *A. caviae* Aero21 and KAM329, respectively (Appendix A). *A. caviae* Aero21 was from hospital wastewater in Brazil [54] and KAM329 from an unknown source in Japan. The next most similar *Aeromonas* isolates had ≥97.65% sequence identity within the *trb* operon and were found on chromosomes from isolates (*Aeromonas* sp. ASNIH1, ASNIH5, and ASNIH7) in the *bla*_KPC_ study mentioned above (Appendix A). *Aeromonas* sp. ASNIH5 and ASNIH7 were acquired from wastewater manholes outside the hospital associated with the patient infected with *Aeromonas* ASNIH1 in 2015. Together, this suggests a correlation of *trb* operon genes and carbapenemases among *Aeromonas* species.

Considering the diverse environments *Aeromonas* is known to colonize, the complex mobilome associated with this genus is ever evolving. It is evident that *Aeromonas* engages in interspecies horizontal gene transfer [55], including with members of *Enterobacterales* [56]. Additionally, we found MGEs scattered throughout pMAH4, including an IS6100 composite transposon between 30,877–53,665 bp (Figure 3 and Appendix A). Within this composite transposon is a type 1 integrase as well as multiple ARGs. The presence of IS26 and ISCfr1 inside this composite transposon likely reflects numerous mobilization events that resulted in this multi-drug-resistant plasmid. Additionally, ISPst3, ISAs1, and ISAhy2 are all within 5000 bp of each other (Appendix A), reaffirming a high frequency of recombination and HGT events in this isolate.

The presence of SFO-1 from *Aeromonas* isolates collected from humans, wastewater, and dogs further supports addressing antimicrobial resistance from a One Health approach. In previous work, we used *Aeromonas* as a global indicator species for analyzing antimicrobial resistance over a twenty-year period. We identified similar resistance levels across human, agricultural, and environmental sectors for the majority of the 21 antimicrobials investigated; the few exceptions were mainly for last-resort antimicrobials like cefepime and aztreonam [3]. Among these antibiotics, wastewater contains significantly higher resistance profiles than all other sectors, which stresses that wastewater may be an origin or source for increased HGT and ARG acquisition. The findings of SFO-1 in wastewater globally further highlight the importance of characterizing AMR from multiple sectors so that the ultimate drivers of resistance mobilization and proliferation may be identified. Lower socio-economic countries are commonly associated with increased levels of AMR through all sectors [3], which are also correlated frequently with antimicrobial overuse and an absence or deficiency of wastewater treatment facilities [3,57]. It is imperative to develop global policies regulating antimicrobial use and for higher socio-economic countries to increase support for infrastructure, like wastewater systems, in countries with lower gross domestic income.

## 4. Conclusions

We report the first clinical isolate of an *Aeromonas* species encoding *bla*_SFO-1_ from a multi-drug-resistant strain in the USA. Isolate MAH-4 contained numerous ESBLs both chromosomally and on a 327 kb plasmid. Genomic evidence strongly indicates interspecies HGT between *Klebsiella* and *Aeromonas*. This conclusion is further strengthened when taking into account the common presence and relatively high abundance of both genera in wastewater. Wastewater may be an accelerator of AMR as it often harbors compounds like heavy metals and subinhibitory concentrations of antimicrobials, both of which have been shown to increase genetic exchange. Although wastewater presents a difficult challenge in the fight against AMR, it also provides an opportunity via wastewater-based epidemiology to monitor rising resistance patterns and ARGs across disparate areas so that resistance risks can be identified before they become clinical problems.

## Figures and Tables

**Figure 1 microorganisms-12-00494-f001:**
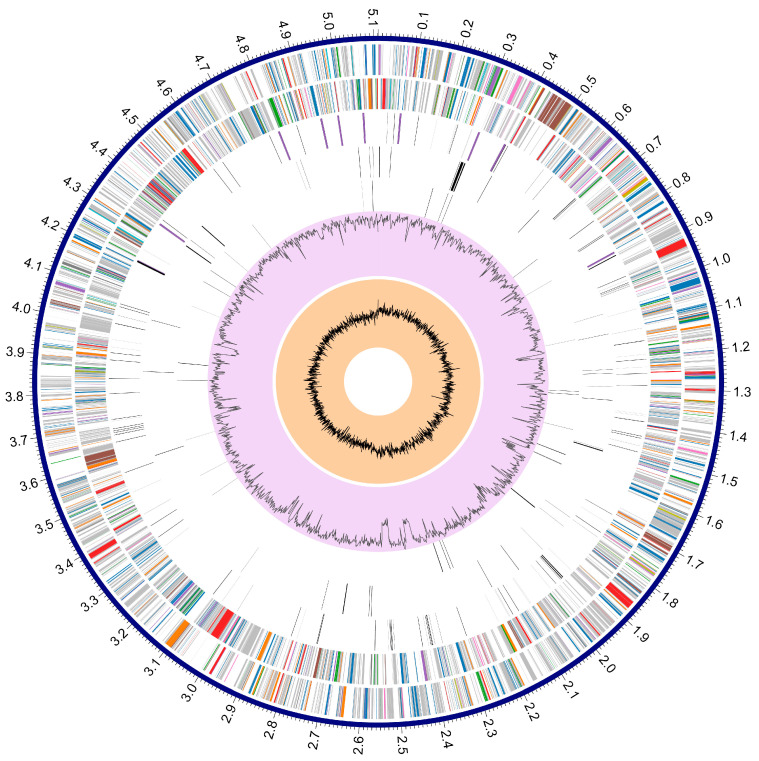
Chromosome map of *Aeromonas hydrophila* MAH-4. The coding sequences (CDS) of the forward strand are on the outermost ring followed by the CDS of the reverse strand. The colors of the CDS represent the various subsystems: blue-metabolism; green-protein processing; orange-energy; purple-stress response; defense, and virulence; red-membrane transport; pink-DNA processing; brown-cellular processes; gray-RNA processing; olive-cell envelope; artic blue-miscellaneous; turquoise-regulation and cell signaling. The third ring from the outside represents RNA genes. The fourth ring represents antimicrobial resistance CDS followed by virulence CDS on the fifth ring. The purple ring presents the GC content throughout the chromosome followed by the GC skew in the innermost ring.

**Figure 2 microorganisms-12-00494-f002:**
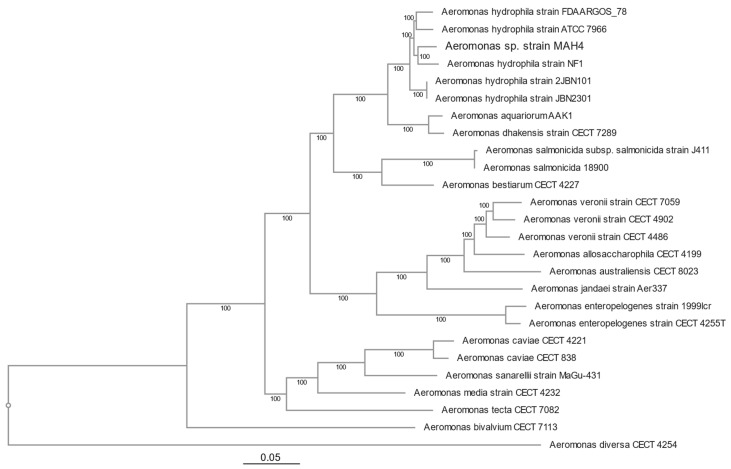
Phylogenetic tree reconstruction derived from conserved concatenated genes of clinical isolate *Aeromonas hydrophila* MAH-4 and other representative *Aeromonas* species.

**Figure 3 microorganisms-12-00494-f003:**
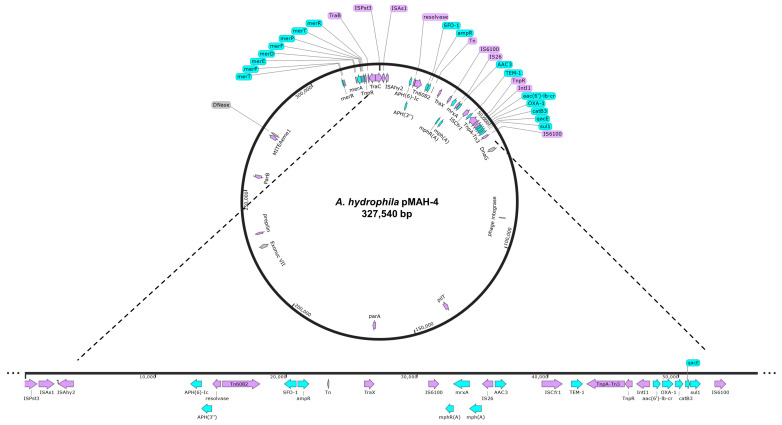
Mobile genetic element (MGE) and antimicrobial resistance gene (ARG) characterization of pMAH4. Blue arrows represent predicted ARGs with purple arrows representing MGEs of the 327,540 bp plasmid from *Aeromonas hydrophila* MAH-4. SnapGene v. 7.1.1 was used to create the figure.

**Figure 4 microorganisms-12-00494-f004:**
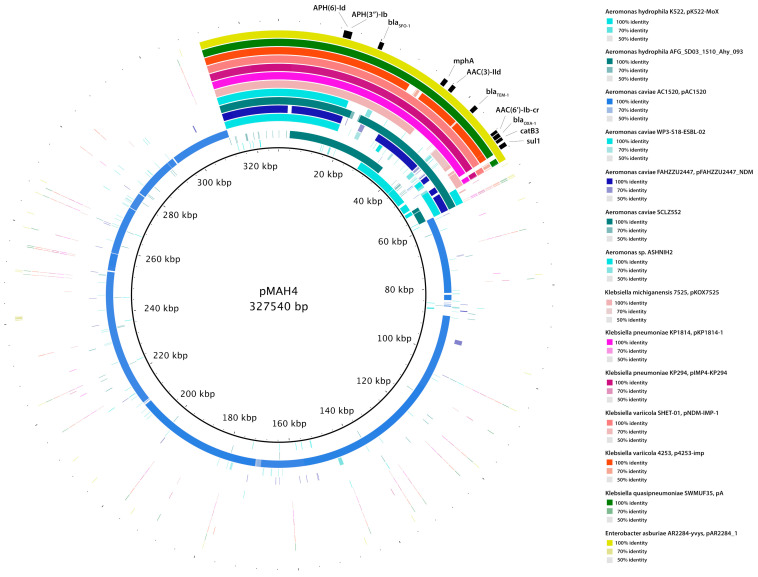
Ring image indicating the sequence identity via blastn analysis of pMAH4 to other bacterial genome sequences (plasmids). The assembled pMAH4 sequence is indicated by the innermost ring. Antimicrobial resistance genes associated with pMAH4 are identified and mapped in black in the outer ring. Internal colored rings indicate the blastn sequence matches (>50% identity) of each individual bacterial isolate plasmid to pMAH4. Ring color shading delineates different levels of sequence identity from 50–100%. The genome sequences included in the comparison, from innermost to outermost ring, are: *Aeromonas hydrophila* K522 (CP118701), *Aeromonas hydrophila* AFG-SD03_1510_Ahy_093 (PUTQ01000030), *Aeromonas caviae* AC1520 (CP120943), *Aeromonas caviae* WP3-S18-ESBL-02 (AP022013), *Aeromonas caviae* FAHZZU2447 (CP100393), *Aeromonas caviae* SCLZS52 (CP091176), *Aeromonas* sp. ASHNIH2 (CP026406), *Klebsiella michiganensis* 7525 (CP065475), *Klebsiella pneumoniae* KP1814 (KX839207), *Klebsiella pneumoniae* KP294 (CP083446), *Klebsiella variicola* SHET-01 (CP050681), *Klebsiella variicola* 4253 (CP135069), *Klebsiella quasipneumoniae* SWMUF35 (CP068445), *Enterobacter asburiae* AR2284-yvys (CP083831).

**Table 1 microorganisms-12-00494-t001:** Phenotypes and genotypes of antimicrobial resistance of *Aeromonas hydrophila* MAH-4.

Antibiotic		Resistance/ Susceptibility	Disk Diffusion	Resistance/ Susceptibility	Potential ARG Correlation
	MIC	(R/S)	(mm)	(R/S)	
Amikacin	<8	S			*aac6′-lb-cr*
Amoxicillin–clavulanic acid		R			*bla_OXA-1_*
Ampicillin	>16	R			*ampH*, *bla_SFO-1_*, *bla_TEM-1_*, *bla_OXA-1_*
Aztreonam	>16	R	6	R	*bla_SFO-/ampC_*
Cefazolin	>16	R			*bla_TEM-1_*
Cefepime	>16	R			*bla_OXA-1/SFO-1_*
Cefoxitin	>16	R			*bla_SFO-1_*
Ceftazidime	>16	R			*bla_SFO-1/ampC_*
Ceftazidime/avibactam	1/4				
Cefotaxime			6	R	*bla_SFO-1/ampC_*
Ceftriaxone	>32	R			*bla_SFO-1_*
Cefuroxime	>16	R			*bla_SFO-1/ampC_*
Chloramphenicol			18	S	*catB3*
Ciprofloxacin	>2	R	6	R	*aac6′-lb-cr*
Ertapenem	>2	R			*cphA2*
Meropenem			24	S	
Gentamicin	>8	R			*aac(3)-IId*
Levofloxacin	>4	R			
Minocycline	2				
Moxifloxacin	> 4	R			*aac6′-lb-cr*
Piperacillin–tazobactam	>64/4	R			*bla_OXA-1_*
Tetracycline	≤2	S	15	S	
Tobramycin	>8				*aac6′-lb-cr*, *aac(3)-IId*
Trimethoprim–sulfamethoxazole	1/19	S	14	I	*sul-1*
*Antibiotic susceptibility not performed*					
Erythromycin					*mph(A)*
Streptomycin					*aph(3″)Ib*, *aph(6)-Id*

**Table 2 microorganisms-12-00494-t002:** Similarity and location of ARGs from *A. hydrophila* MAH-4.

Drug Class		AA Similarity	% Length of Reference	ARO Reference/Sequence ID	Reference Species
**Chromosomal-encoded**					
Beta-lactamases	*OXA-726 (ampH)*	262/264	100	WP_016352393.1	*A. hydrophila*
	*cphA2*	252/254	100	WP_323883209	*A. hydrophila*
	*FOX/MOX (ampC)*	375/382	100	WP_323974820.1	*A. hydrophila*
**Plasmid-encoded**					
Beta-lactamases	*SFO-1*	295/295	100	30069866	*Enterobacter cloacae*
	*TEM-1*	286/286	100	3000873	*Salmonella enterica*
	*OXA-1*	276/276	100	3001396	*Klebsiella pneumoniae*
Macrolides	*mphA*	301/301	100	3000316	*Escherichia coli*
Aminoglycosides	*aac(3)-IId,*	286/286	100	3004632	*Escherichia coli*
	*aac(6* *′)-lb-cr6*	197/199	100	3005116	*Escherichia coli*
	*aph(6)-Id*	277/278	100	3002660	*Pseudomonas aeruginosa*
	*aph(3* *″)-lb*	265/267	100	3002639	*Pseudomonas aeruginosa*
Phenicols	*catB3*	210/210	100	30002676	*Enterobacter cloacae*
Sulfonamides	*sul1*	279/279	100	3000410	*Vibrio fluvialis*

## Data Availability

Data are contained within the article and Appendix A.

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
