# Peer review of "First Report and Characterization of a Plasmid-Encoded blaSFO-1 in a Multi-Drug-Resistant Aeromonas hydrophila Clinical Isolate"

_microorganisms, 2024, doi:10.3390/microorganisms12030494_

Round 1

Reviewer 1 Report

Comments and Suggestions for Authors

It’s not often that perfect articles like this one First report and characterization of a plasmid-encoded blaSFO-1 in a multi-drug resistant Aeromonas hydrophila clinical isolate come for review. The work is written in clear scientific language. There is no doubt about the English language. Everything's great. The article is devoted to the first clinical case of plasmid-encoded blaSFO-1 in the multidrug-resistant bacterium Aeromonas Hydrophila. Chromosome maps are just a dream for any laboratory) I don’t see any obstacles for the article to be published already. I would be interested to know, as a recommendation: from which patient was the abdominal fluid obtained?

Author Response

We would like to thank the reviewer for the kind comments. Regarding the patient, very interesting question . Unfortunately we only had minimal information but agree on the interest level

Reviewer 2 Report

Comments and Suggestions for Authors

Line 22 I am assuming ARG stands for antibiotic resistance genes – this should be defined.

Line 49 states: “While these ESBL types constitute the majority of ARGs within common pathogens”. I would contest this statement. TEM-1 which gives rise to amoxicillin resistance is not an ESBL and is much more common than any ESBL enzyme. Please revise the sentence.

Line 56 and 57: correct the spelling of E. cloacae.

Table 1. TEM-1 does not normally give rise to resistance to aztreonam or cefazolin. Please revise or explain. Could the resistance to aztreonam and some of the cephalosporins be related to de-repressed AmpC activity?

Table 1: Are the authors confident that aac6 enzymes causing high level resistance to quinolones?

What was the susceptibility of this isolate to meropenem?

Line 231: A  a sulfonamide resistant dihydropteroate synthase was found. Is this consistent with susceptibility to co-trimoxazole ? (line 144)

Line 345: I think the conclusions section should be more succinct and concentrate on the findings of this study. It currently contains elements of discussion e.g. the comment about possible misidentification as CTX-M is not a conclusion. It is unusual to cite references in the conclusions.

Author Response

We appreciate the reviewer’s thorough review and comments. Together, they have made this a stronger manuscript. Our comments are provided in purple and italics below.  

Line 22 I am assuming ARG stands for antibiotic resistance genes – this should be defined.

This has been spelt out in new draft

Line 49 states: “While these ESBL types constitute the majority of ARGs within common pathogens”. I would contest this statement. TEM-1 which gives rise to amoxicillin resistance is not an ESBL and is much more common than any ESBL enzyme. Please revise the sentence.

We appreciate the reviewer’s point and agree TEM-1 is not an ESBL. The sentence was generalizing the ESBL families. However, to address the reviewer’s comment, we have restated for clarification, as well as line 46.  

Line 56 and 57: correct the spelling of E. cloacae.

Thank you for catching this and changed.

Table 1. TEM-1 does not normally give rise to resistance to aztreonam or cefazolin. Please revise or explain. Could the resistance to aztreonam and some of the cephalosporins be related to de-repressed AmpC activity?

We have updated Table 1 to address the reviewer’s comment, and agree regarding the resistance to aztreonam could be due to derepressed ampC activity and SFO-1 [1] as well. Additionally, we have included a few sentences discussing the variety of efflux pumps found within the chromosome to support unexplained resistance patterns (lines 173-178).

Table 1: Are the authors confident that aac6 enzymes causing high level resistance to quinolones?

It does not produce high level resistance, but rather moderate resistance to ciprofloxacin and other quinolones. We have modified the text in line 240 to state this and clarified the table to only have aac6 associated with ciprofloxacin considering the resistance is not seen with quinolones lacking piperazinyl substituent and the associated secondary amino nitrogen N4 [2] . Considering we are not aware of other mechanisms providing resistance to ciprofloxacin, we put in the only ARG in our genome that might be responsible. Chromosomal mutations and qnr were not evident surprisingly. We did include a statement about the role of efflux pumps that might be adding to resistance to these quinolones as stated above.

What was the susceptibility of this isolate to meropenem?

We have included disk diffusion assay, which was performed in the research lab in contrast to the clinical lab. An additional section in the methods was included as well to discuss the assay (line 100-105)

Line 231: A  a sulfonamide resistant dihydropteroate synthase was found. Is this consistent with susceptibility to co-trimoxazole ? (line 144)

It was a surprise to us as well considering sul1 usually is associated with resistance. We have included a sentence discussing the overlap of qacE delta1 and sul1, which might impact its expression (lines 251-253) [3]. We have also included disk diffusion data which identified intermediate resistance, supporting its presence at least in low concentrations.

Line 345: I think the conclusions section should be more succinct and concentrate on the findings of this study. It currently contains elements of discussion e.g. the comment about possible misidentification as CTX-M is not a conclusion. It is unusual to cite references in the conclusions.

The conclusion has been modified to address the reviewer’s comment. We have relocated the misidentification sentence to the body (line 214-216) of the manuscript and removed references. We further rearranged the sentences (line 219-223) to transition better into the discussion regarding SFO-1 location in the plasmid.

  1. Matsumoto, Y.; Inoue, M. Characterization of SFO-1, a plasmid-mediated inducible class A beta-lactamase from Enterobacter cloacae. Antimicrob Agents Chemother 1999, 43, 307-313, doi:10.1128/aac.43.2.307.
  2. Rodriguez-Martinez, J.M.; Machuca, J.; Cano, M.E.; Calvo, J.; Martinez-Martinez, L.; Pascual, A. Plasmid-mediated quinolone resistance: Two decades on. Drug Resist Updat 2016, 29, 13-29, doi:10.1016/j.drup.2016.09.001.
  3. Johnson, Z.I.; Chisholm, S.W. Properties of overlapping genes are conserved across microbial genomes. Genome Res 2004, 14, 2268-2272, doi:10.1101/gr.2433104.
